# Community-CL: An Enhanced Community Detection Algorithm Based on Contrastive Learning

**DOI:** 10.3390/e25060864

**Published:** 2023-05-29

**Authors:** Zhaoci Huang, Wenzhe Xu, Xinjian Zhuo

**Affiliations:** School of Science, Beijing University of Posts and Telecommunications, Beijing 100876, China; zc.huang@bupt.edu.cn (Z.H.); zhuoxj@bupt.edu.cn (X.Z.)

**Keywords:** community detection, contrastive learning, graph neural network

## Abstract

Graph contrastive learning (GCL) has gained considerable attention as a self-supervised learning technique that has been successfully employed in various applications, such as node classification, node clustering, and link prediction. Despite its achievements, GCL has limited exploration of the community structure of graphs. This paper presents a novel online framework called Community Contrastive Learning (Community-CL) for simultaneously learning node representations and detecting communities in a network. The proposed method employs contrastive learning to minimize the difference in the latent representations of nodes and communities in different graph views. To achieve this, learnable graph augmentation views using a graph auto-encoder (GAE) are proposed, followed by a shared encoder that learns the feature matrix of the original graph and augmentation views. This joint contrastive framework enables more accurate representation learning of the network and results in more expressive embeddings than traditional community detection algorithms that solely optimize for community structure. Experimental results demonstrate that Community-CL achieves superior performance compared to state-of-the-art baselines in community detection. Specifically, the NMI of Community-CL is reported to be 0.714 (0.551) on the Amazon-Photo (Amazon-Computers) dataset, which represents a performance improvement of up to 16% compared with the best baseline.

## 1. Introduction

Community detection is a crucial problem in network analysis that aims to identify groups of nodes that are more interconnected with each other than with the rest of the network [1]. It has a wide range of applications, including social network analysis [2], recommendation systems [3], and epidemiology [4], among others. Specifically, ref. [5] presents a novel perspective by exploring the community evolutions in TikTok’s dangerous and non-dangerous challenges, providing valuable insights for community detection in social media. And ref. [6] utilize a community detection model to analyze text streams from microblogging sites, aiming to detect users’ interest communities. Over the years, numerous community detection algorithms have been proposed, ranging from traditional methods based on modularity optimization [7] to more recent ones using deep learning techniques [8]. Despite the great progress achieved so far, most existing community detection methods rely on labeled data, which is often costly and time-consuming to obtain. In contrast, unsupervised community detection methods [9], which do not require any prior knowledge about the network structure, have received less attention in the literature.

One promising approach to address this lack is contrastive learning [10], which learns the general features of a dataset without the need for manual labels or annotations. It involves learning a similarity metric between pairs of samples and optimizing a loss function that encourages same-class samples to be close to each other while being far from samples of other classes. Graph contrastive learning is a recent extension of contrastive learning to network analysis [11], which is typically used to encode the graph structure and generate node or subgraph embeddings. The embeddings are then compared using a contrastive loss function, which encourages similar nodes or subgraphs to be closer to each other in the embedding space than dissimilar ones. Despite its potential, graph contrastive learning for unsupervised community detection is still in its infancy, with relatively few existing methods that specifically target this task.

However, existing methods face several challenges that need to be addressed to improve community detection. One major challenge is the choice of the positive and negative sampling strategy (augmentation method). Unlike in the field of computer vision, where changes to an image such as rotation or distortion do not significantly alter its semantic meaning [12], even minor alterations to the structure of graph data can significantly damage its semantics [13]. Therefore, developing more effective strategies that can capture the complex community structure of the network is an important research direction. Another challenge is the design of the contrastive framework itself. Most existing methods focus on learning a global representation of the network, rather than a node-level or community-level representation that can directly facilitate community detection. Moreover, existing community detection methods often use a two-step learning process [8,14], which can introduce extra errors and lead to a suboptimal result. To address these challenges, it is necessary to explore new ways to incorporate community structure into contrastive learning models and design end-to-end frameworks that can facilitate community detection directly.

To address these issues, we consider the following two aspects.

**Data augmentation**: also referred to as augmented data or augmentation view, involves generating new data from existing data through the random masking or removal of certain elements, such as edges, nodes, or attributes [15]. In this paper, we propose a novel approach for generating augmented data that takes into account the importance of edge structures in communities. Specifically, we use a graph auto-encoder to map the original graph into an embedding space, and a dot product decoder to calculate the probability of edge existence between nodes. By incorporating an augmented-level parameter to filter out low-probability edges, our proposed method generates a reconstructed graph that is more informative, controllable, and preserves more topological information, resulting in more effective data augmentation for community detection tasks.

**Contrastive Framework**: Inspired by CC [16] in computer vision, we argue that the columns of the representation matrix learned by the shared encoder can be considered as the pseudo-label of the community. Therefore, we build a joint contrastive framework in which community-level and node-level contrastive learning are executed synchronously. The joint contrastive framework can increase the similarity of representations of the same community in different views and enhance the consistency of representations of the same node in different views. Meanwhile, the community-level contrastive can achieve end-to-end community detection from an individual clustering perspective by learned community representations.

The main contributions of this paper are summarized as follows:We present a novel end-to-end algorithm for community detection, which leverages a joint contrastive framework to simultaneously learn the community-level and node-level representations.We propose a learnable augmentation view generation scheme that captures the significance of edges in embedding space and generates more informative and diverse augmented data for community detection.We conduct extensive experiments on multiple real-world graph datasets to evaluate the proposed method. The results demonstrate that our approach achieves competitive performance on community detection tasks and the learnable augmentation scheme is effective and robust.

## 2. Related Work

### 2.1. Community Detection

A variety of community-detection algorithms have been developed since the beginning of the 21st century. Traditional methods for community detection are mainly based on modularity theory and clustering [17] algorithms and have made notable progress in recent years [18]. Modularity optimization-based methods are commonly used due to its intuitive and easily applicable characteristics. The Girvan-Newman algorithm evaluates the importance of nodes in the network using betweenness centrality, and then gradually deletes edges in the network according to the size of betweenness centrality, ultimately dividing the network into several communities. The Louvain algorithm [19] uses modularity to evaluate the quality of community partitions, divides the network into several communities, and continuously optimizes modularity until it can no longer be improved. However, these methods may suffer from high computational complexity and poor scalability when dealing with large-scale graphs. In addition, clustering-based methods have also been widely used in the field of community detection, such as K-means [20] and hierarchical clustering [21]. Cluster-based methods usually require setting the number of clusters in advance, so for networks with complex community structures, the results may not be satisfactory.

With the development of Internet technology, deep learning models have gained lots of attention in community detection tasks due to their ability to capture unstructured features and identify high-dimensional nonlinear information [8]. Deep learning techniques for community detection can be broadly categorized into two groups: methods based on Graph Embedding [22] and methods based on Graph Convolutional Networks (GCN) [23]. Graph embedding methods, such as DeepWalk [24], and Node2Vec [25], leverage random walk to maximize the similarity between nodes and neighbourhoods. Methods based on GCN perform convolution operations directly on the graph structure to extract node features and perform community detection [26].

The LGNN algorithm [27] first transforms the original graph into its corresponding line graph and then performs node embedding on the line graph. After the node embedding is completed, the LGNN algorithm uses a linear support vector machine (Linear SVM) to predict labels for each node, thereby achieving supervised community detection. Many other deep learning methods have also been investigated for community detection beyond graph embedding and convolutional networks. One such method is AA-Cluster [28], which states that neighbours of similar nodes and multi-step neighbours should also have similarities. Based on this, hierarchical similarity probabilities are proposed, AA-Cluster uses biased random wandering with Skip-Gram followed by stochastic gradient descent to maximize the co-occurrence probability of similar nodes. Another notable approach is CommunityGAN [29], which employs a generative adversarial network (GAN) [30] for community detection. By generating local neighborhoods of the original graph and using a discriminator network to classify them, CommunityGAN learns the community structure information and produces the final community partitioning results. The aforementioned algorithms are all supervised methods, which heavily rely on labeled data and may suffer from limited generalization ability to handle new or unseen community structures.

Most current unsupervised methods for community detection typically rely on a two-step learning scheme, first learning node embeddings and then using k-means for community segmentation. This approach may lead to an increase in unknown errors. Therefore, developing unsupervised or self-supervised deep learning methods for community detection that can directly learn from graph structures without relying on external labels remains an active research area. As one of the most popular algorithms in the field of self-supervised learning, contrastive learning has provided solutions for many unsupervised tasks. Therefore, this study aims to address the scarcity of unsupervised algorithms for community detection by combining the self-supervised learning strategy of contrastive learning and proposing a new self-supervised community detection algorithm.

### 2.2. Graph Contrastive Learning

Contrastive learning is an unsupervised learning approach that aims to learn the underlying structure of data by comparing pairs of samples. Unlike supervised learning, which requires labeled data, contrastive learning focuses on identifying the similarity or dissimilarity between pairs of samples. As an extension of contrastive learning, graph contrastive learning learns the embedding representations of nodes and edges by utilizing graph data. Unlike traditional graph representation learning methods that rely on supervised information or labels, graph contrastive learning relies solely on contrastive loss functions to train the model, enabling it to learn more robust and useful feature representations in an unsupervised manner. Moreover, graph contrastive learning typically combines data augmentation techniques to improve the performance of the model, enabling it to better deal with issues such as data sparsity and missing values.

The field of graph representation learning has been advancing rapidly in recent years, and contrastive learning has emerged as a promising unsupervised approach for learning representations of nodes and edges in graphs.

DGI [31] is a novel work that applies the contrastive learning framework to graph representation learning and proposes a global-local contrastive framework. In this approach, the graph is corrupted by shuffling features, and the corrupted graph is regarded as the global negative sample. The goal of DGI is to maximise global and local mutual information. In terms of data augmentation for contrastive learning, GraphCL [15] discusses the effects of four different graph data augmentation methods (node dropping, edge perturbation, attribute masking, and subgraph) and their different combinations on graph contrastive learning. The results show that applying augmentation for graph contrastive learning improves the performance of downstream tasks, but the best-fit augmentation strategies vary across different graph datasets. GCA [32] proposed an adaptive augmentation scheme which computed the edge importance from three perspectives: node degree, feature vector and PageRank. They then calculated the probability of edge dropping based on the edge importance. However, the prior probability of edge importance is not learnable. MVGRL [33] and GCC [34] apply subgraph augmentation and contrastive. MVGRL uses a diffusion kernel to add edges and then uses subgraph sampling to obtain contrastive pairs; while GCC constructs a subgraph from the ego-network generated by the nodes. The subgraphs generated from different nodes or graphs can be considered as negative sample pairs. However, all the above methods rely on various intuition-based augmentation strategies to transform the graph structure. This paper intends to contribute to the field of graph representation learning by introducing a novel graph augmentation scheme and contrastive framework specifically for community detection.

## 3. Methods

As shown in Figure 1, our proposed framework combines the power of graph autoencoders, graph convolutional neural networks, and contrastive learning to enhance the topological structure and feature information of a graph, leading to improved performance on downstream tasks. Figure 1a is an augmentation generator, generated based on a graph autoencoder. It is superior to random edge drop methods as it preserves more topological structure and feature information of the original graph by learning a low-dimensional embedding representation. An importance threshold is employed to regulate the augmented level, which provides control over the diversity and richness of the generated augmented graph. In contrast, random edge missing methods lack this controllability. Figure 1b is a shared encoder, which utilizes a two-layer GCN to aggregate information from the graph’s neighborhoods more effectively. This approach enables the model to capture local and global structural information and enhance the overall performance of downstream tasks.

Two contrastive heads are designed in Figure 1c, one for node-level contrastive learning and the other for community-level contrastive learning. This design allows the model to learn both fine-grained node representations and high-level community representations, which can capture different levels of graph structure and enhance the overall performance of downstream tasks. The node-level contrastive head focuses on learning node-level representations by contrasting positive pairs of augmented views of the same node and negative pairs of different nodes. This allows the model to capture the local structural information and the context of each node. The community-level contrastive head, on the other hand, learns representations for communities by contrasting positive pairs of augmented views of the same community and negative pairs of different communities. This allows the model to capture global structural information and interdependencies between nodes. By combining these two contrastive heads, the proposed method can learn both fine-grained and high-level representations of the graph, and the soft labels generated by the community-level contrastive head can be utilized to obtain the community detection results for each node directly.

### 3.1. Generator

Given an original graph G=(V,E), where *V* and *E* are the nodes set and edges set, respectively, we denote the adjacency matrix of the graph as $A\in {\left\{0,1\right\}}^{n\times n}$ and the nodes’ feature matrix as $X\in {\left\{0,1\right\}}^{n\times d}$, where n=|V| is the number of nodes and *d* is the dimension of features. A two-layer graph convolutional neural network (GCN) is first used to aggregate the neighborhood information of the nodes and generate an embedding representation of the nodes. The adjacency matrix and the feature matrix are multiplied to achieve the convolution operation on the graph structure, which captures the structural information of the graph. In the GCN model, the same adjacency matrix is used at each layer to achieve information sharing and propagation, as shown in Equation (Equation 1):(1)$$Z=\mathrm{GCN}\left(X,A\right)=\hat{A}\mathrm{ReLu}\left(\hat{A}XW_{0}\right)W_{1},$$
where $W_{0}$, $W_{1}$ are the weight matrix that need to be learned, Relu is a nonlinear activation function that serves to perform a nonlinear transformation of the input to enhance the expressiveness of the model, $\hat{A}=D^{\frac{1}{2}}AD^{\frac{1}{2}}$, and *D* is the degree matrix of the graph. If the graph does not have a feature matrix, a unit matrix $I\in {\left\{0,1\right\}}^{n\times n}$ can be used to replace the *X*. A dot product decoder can be applied to achieve the reconstructed adjacency matrix $\tilde{A}$,
(2)$$\tilde{A}=\mathrm{sigmoid}\left(ZZ^{T}\right).$$ The cross-entropy is used as a loss function by maximizing the similarity between the original matrix and the reconstructed adjacency matrix.
(3)$$L=-\frac{1}{n}\sum_{i,j\in E}(a\log(\tilde{a_{ij}}+\left(1-a_{ij}\right)\log\left(1-{\tilde{a}}_{ij}\right),$$
where $a_{ij}$ is the element of the original adjacency matrix, and ${\tilde{a}}_{ij}$ is one of the reconstructed adjacency matrices. ${\tilde{a}}_{ij}$ can be seen as the existence probability of edges between node *i* and node *j* of the reconstructed graph. The element $a_{ij}$ in *A* generated after pre-training can be regarded as the probability of the existence of concatenated edges between node *i* and node *j* in the reconstructed adjacency matrix.

We have established an augmentation level $\tau_{a}\in \left[0,5\right]$ to regulate the trade-off between diversity and precision of the generated augmentation view. By varying the augmentation level, we can regulate the threshold probability for edge retention in the generated augmented graph,
(4)$$\tilde{A}=\left\{\begin{array}{cc} 1 & {\tilde{a}}_{ij}>\mathrm{sigmoid}\left(\tau_{a}\right)\\ 0 & {\tilde{a}}_{ij}\leq \mathrm{sigmoid}\left(\tau_{a}\right). \end{array}\right.$$
where the importance threshold is computed by the following function:(5)$$\mathrm{sigmoid}\left(\tau_{a}\right)=\frac{1}{1+e^{\tau_{a}}}.$$

The converted $\tilde{A}$ will be used as input data for the contrastive model. $\tau_{a}$ and the importance threshold preserve the sparsity of generated view, which reduces the computational complexity and makes the algorithm still efficient on the large graph. And the training strategy of the generator is described in Algorithm 1.
**Algorithm 1** The framework of generator**Require:** Original Graph *G*, Augmented level $\tau_{a}$, Training Epoch $E_{1}$ and Structure of $g_{\theta }$**Ensure:** Augmentation View $G^{T}$  1:**for** epoch = 1 to $E_{1}$ **do**  2: Encode the original G by $Z=g_{\theta }\left(G\right)$  3: Decode *Z* by a dot product decoder using Equation (Equation 2)  4: Compute Loss *L* using Equation (Equation 3)  5: Update $g_{\theta }$ through gradient descent to minimize *L*  6:**end for**  7:Generate reconstruct adjacency matrix $\tilde{A}$ using the augmented level $\tau_{a}$ (see Equation (Equation 4)).

### 3.2. Shared Graph Convolution Encoder

After generating the augmented views, we employ a shared graph convolutional neural network to learn the representation matrices of the original graph and the augmented views. The learned representation is denoted as $H^{ori}$ and $H^{aug}$. GCN can effectively capture the structural information and local features between nodes, enabling accurate modeling of similarity in contrastive learning. By leveraging the advantages of GCN, we aim to enhance the performance of our proposed method in the contrastive learning task.

### 3.3. Node-Level Constrastive Head

The purpose of contrastive learning is to learn a low-dimensional embedding representation in which positive sample pairs are similar, and negative sample pairs are distinct. We use pseudo-labels based on the original graph and the augmentation view to construct positive and negative sample pairs. Each original graph and its augmented graph contain n nodes, resulting in a total of 2n nodes. For a specific node in an original graph, its corresponding node in the augmented graph is considered a positive sample pair, while all other 2n − 2 nodes in the original and augmented graphs are considered negative sample pairs. To alleviate information loss caused by contrastive learning, a two-layer nonlinear MLP encoder $g_{N}ode\left(\cdot \right)$ is used to map the feature matrix obtained by the shared encoder to a low-dimension embedding space. The embeddings for the original and augmented graphs are denoted as $Z_{Node}^{ori}=g_{Node}\left(H^{ori}\right)$ and $Z_{node}^{aug}=g_{Node}\left(H^{aug}\right)$, respectively. The $z_{i}^{ori}$ and $z_{i}^{aug}$ are feature vectors of node *i* in different views. We compute the similarity between all pairs of nodes using the following cosine similarity,
(6)$$s\left(z_{i}^{ori},z_{j}^{aug}\right)=\frac{\left(z_{i}^{ori}\right){\left(z_{j}^{aug}\right)}^{T}}{∥z_{i}^{ori}∥∥z_{j}^{aug}∥},$$
where i,j∈[1,n]. To optimize the similarity of node pairs, a loss function is utilized. Specifically, for a given node *i* of the original graph, the associated pairwise loss is computed as follows in Equation (Equation 7):(7)$$l_{i}^{ori}=-\log\frac{\exp\left(s\left(z_{i}^{ori},z_{i}^{aug}\right)\right)/\tau_{Node}}{\sum_{j=1}^{n}\left[\frac{\exp(s\left(z_{i}^{ori},z_{i}^{aug}\right))}{\tau_{Node}}+\frac{\exp(s\left(z_{i}^{ori},z_{j}^{aug}\right))}{\tau_{Node}}\right]},$$
where $\tau_{Node}$ is the temperature parameter that controls the sharpness of the probability distribution. The numerator of the loss function represents the similarity of the positive sample pair, while the denominator corresponds to the sum of the similarities between the given node and all others in the graph. $s(z_{i}^{ori},z_{j}^{aug})$ computes the similarity of node *i* and non-corresponding node *j*. To regulate the model’s sensitivity to similarities, $\tau_{Node}$ is introduced. By adjusting the value of $\tau_{Node}$, we can balance the robustness and discriminability of the model [35].

At the node level, the objective of the node-level contrastive head is to enhance the similarity between each original node and its corresponding augmented node. Therefore, we compute the contrastive loss at the node level for each node in the original graph and its augmentation view. The contrastive loss between the *i*-th augmented node and its corresponding original node is denoted as $l_{i}^{aug}$, computation of $l_{i}^{aug}$ is same as $l_{i}^{ori}$ introduced in Equation (Equation 6). Therefore, the overall contrastive loss can be represented as
(8)$$Loss_{Node}=\frac{1}{2n}\sum_{i=1}^{n}\left(l_{i}^{ori}+l_{i}^{aug}\right).$$

### 3.4. Community-Level Contrastive Head

We project the feature representation $H^{ori}$ and $H^{aug}$ into the latent space of the community dimension using another two-layer MLP encoder $g_{Comm}\left(\cdot \right)$. This results in community embedding matrix $Z_{Comm}^{ori}\in R^{n\times M}$ and $Z_{Comm}^{aug}\in R^{n\times M}$, where *M* is the number of communities. In other words, $Z_{Comm}^{ori}$ and $Z_{Comm}^{aug}$ can be viewed as feature matrices in the community dimension, where the *i*th row of the matrix represents the feature vector for node *i* and the element of $z_{ip}$ represents the probability that node *i* belongs to the community *p*. However, if we transpose the feature matrix $Z_{Comm}^{k}\in R^{n\times M}$ to ${Z_{Comm}^{k}}^{T}\in R^{M\times n}$, where k∈{ori,aug}, the *p*th row can be seen as the feature vector of community *p*, where each element $y_{j}i$ represents the probability that community *p* contains node *i*. We denoted the feature vector of community *p* as $y_{p}^{k}$.

The cosine similarity of the community pairs is calculated in the community embedding space
(9)$$s\left(y_{p}^{ori},y_{q}^{aug}\right)=\frac{\left(y_{p}^{ori}\right)\left({y_{q}^{ori}}^{T}\right)}{∥y_{p}^{ori}∥∥y_{q}^{aug}∥},$$
where p,q∈[1,M]. We set the community-level temperature parameter as $\tau_{Comm}$, and use the normalized temperature-scale cross-entropy loss function to optimize the community contrastive pairs. This helps to ensure that for each community feature vector $y_{p}^{ori}$, its similarity is maximized with its corresponding community feature vector $y_{p}^{aug}$, while its similarity to non-corresponding communities is minimized.
(10)$${\hat{l}}_{p}^{ori}=-\log\frac{\exp\left(s\left(y_{p}^{ori},y_{p}^{aug}\right)\right)/\tau_{Comm}}{\sum_{p=1}^{M}\left[\frac{\exp(s\left(y_{p}^{ori},y_{p}^{aug}\right))}{\tau_{Comm}}+\frac{\exp(s\left(y_{p}^{ori},y_{q}^{aug}\right))}{\tau_{Comm}}\right]}.$$ Similarity with the node contrastive loss, the contrastive loss between the *p*-th augmented community and its corresponding original community is denoted as ${\hat{l}}_{p}^{aug}$, computation of ${\hat{l}}_{p}^{aug}$ is the same as ${\hat{l}}^{ori}pi$, introduced in Equation (Equation 10). After traversing the contrastive loss of all communities, a penalty term is introduced to mitigate the issue of the majority of nodes being assigned to the same community. The assignment probability $P(y_{q}^{k})$ is calculated for each community:(11)$$P\left(y_{q}^{k}\right)=\frac{p(y_{qi}^{k})}{\sum_{q=1}^{M}\sum_{i=1}^{N}p\left(y_{qi}^{k}\right)},$$
where k∈{ori,aug}. To calculate the sum of the entropy of community assignment probabilities of the original and augmentation views, I(Y):(12)$$I\left(Y\right)=-\sum_{i=1}^{M}[P(y_{i}^{ori}\log P\left(y_{i}^{ori}\right)+P\left(y_{i}^{aug}\log P\left(y_{i}^{aug}\right)\right].$$ If all the nodes are assigned to the same community, the community assignment probabilities will be highly imbalanced and the entropy of community assignment will be very low, resulting in a higher penalty term. In contrast, if each node is equally likely to be assigned to any community, the entropy of community assignment will be maximized, resulting in a lower penalty term. The final community-level loss is computed as:(13)$$Loss_{Comm}=\frac{1}{2M}\sum_{p=1}^{M}\left({\hat{l}}_{p}^{ori}+{\hat{l}}_{p}^{aug}\right)-I\left(Y\right).$$

### 3.5. Object Function

Our proposed method differs from traditional graph clustering approaches, which often focus solely on either global or local information. Instead, we combine both community-level and node-level heads, which allows for a more comprehensive understanding of the graph and more informed clustering decisions. By optimizing the node-level and community-level contrastive heads simultaneously in a single-stage, end-to-end process, our model can effectively leverage both types of information to achieve superior clustering results. We introduce an overall objective function to comprise node-level and community-level contrastive losses, i.e.,
(14)$$L=\alpha Loss_{n}+\left(1-\alpha \right)Loss_{c}.$$ Here, the hyper-parameter α is used to balance the node-level and community-level contrastive losses. Through experimentation, we found that setting α=0.5 works well. Our proposed model’s complete training and testing process is presented in Algorithm 2. After the training process is completed, the argmax function is used to aggregate the embeddings of each community, resulting in the final assignment of each node to a specific community. Specifically, the final community representation *H* is computed by an argmax function, which returns the index of the largest element in the vector. The returned result *c* is the assignment vector of nodes.
**Algorithm 2** The framework of contrastive learning**Require:** Original Graph, *G*, Augmentation View, $G^{T}$, Training Epoch, $E_{2}$, Temperature parameter, $\tau_{N},\tau_{C}$, Community number, *M*, Structure of *f*, $g_{Node}$, and $g_{Comm}$.**Ensure:** Community assignments.  1:**for** epoch = 1 to $E_{2}$ **do**  2: compute node and community representations by H=f(G), $\tilde{H}=f\left(G^{T}\right)$ $z_{i}^{ori}=g_{Node}\left(h_{i}^{ori}\right)$, $z_{i}^{aug}=g_{Node}\left(h_{i}^{aug}\right)$ $y_{p}^{ori}=g_{Comm}\left(h_{p}^{ori}\right)$, $y_{p}^{aug}=g_{Comm}\left(h_{p}^{aug}\right)$  3: Compute node-level contrastive Loss $L_{N}$, community-level contrastive Loss $L_{C}$ and overall loss *L*  4: Update *f*, $g_{Node}$, and $g_{Comm}$ through gradient descent to minimize *L*  5:**end for**  6:Extract feature by H=f(G)  7:Compute community assignment by $c=\mathrm{argmax}g_{C}\left(h\right)$

## 4. Experiments

### 4.1. Experimental Setup

#### 4.1.1. Datasets

We evaluate our proposed method on six different graph datasets of varying sizes, including two small datasets, Cora & Citeseer, and four middle-sized datasets Amazon-Photo & Amazon-Computers, Coauthor-CS, and WikiCS. The detailed statistics are listed in Table 1.

**Cora & Citeseer** [36] are two citation networks where nodes represent papers and edges are established if two papers have a reference relationship. The papers are described using a bag-of-words, and the one-hot bag-of-words is used as the feature for the network. In Cora and Citeseer, the nodes are assigned to seven and six classes, respectively, based on the papers’ type.

**Amazon-Photo & Amazon-Computers** [37] are two co-purchase networks collected by crawling the Amazon website. The nodes represent the products, and if two products are purchased together, they have an edge. The features are bag-of-words vectors extracted from product reviews, and the class labels are given by the product category.

**Coauthor-CS & Coauthor-Physics** [37] are co-authorship graphs based on the Microsoft Academic Graph from the KDD Cup 2016 challenge. The nodes represent authors, and they are connected by an edge if they co-authored a paper. The features represent paper keywords for each author’s papers, and class labels indicate each author’s most active fields of study.

**WikiCS** [38] is constructed based on the Computer Science articles in Wikipedia. The network consists of nodes corresponding to articles, with edges based on hyperlinks. There are 10 classes representing different branches of the field.

#### 4.1.2. Implementation Details

To compare our proposed method with previous work, we use a 2-layer GCN network as the shared encoder in all experiments, as well as in the primary network of the baseline method. We use the generator described in Section 3.1 to construct the augmentation view. The hidden layer dimension of the generator is set to 16, and the initial learning rate is set to 0.001. We use a dot product decoder to get the reconstructed probability matrix after 200 training epochs. For all the datasets, we set the augmented level to 5. The shared encoder encodes the input original graph and augmentation view to generate 128-dimensional feature vectors for subsequent comparison learning. The temperature parameters $\tau_{N}$, and $\tau_{C}$ are set to 0.5 and 1, respectively.

The augmentation view generator and contrastive model both are optimized by Adam Optimizer [39]. Adam is a variant of stochastic gradient descent (SGD) that prevents the learning rate from becoming too large when the gradient is large, ensuring the stability of parameter values. This optimizer has been widely used and proven effective since it was proposed.

We employ PyTorch and PyG (PyTorch Geometric Library) to download and process the graph data. The experiments are conducted using the Nvidia GeForce RTX 3080Ti GPU.

#### 4.1.3. Evaluation Metrics

The two metrics we use to evaluate our method are Normalized Mutual Information (NMI) and Adjusted Rand Index (ARI). These metrics are commonly used in evaluating clustering algorithms, with higher values indicating better clustering results for all metrics.

#### 4.1.4. Leading Example

In order to enhance the comprehensibility of our experiments, we present a leading example as follows:Step 1: We downloaded the Cora dataset using PyG. The Cora dataset consists of 2708 nodes, 5429 edges, and a total of 7 categories. The dataset includes 1433 features, each of which is represented by only 0/1. Thus, we obtained the adjacency matrix $A\in R^{2708\times 2708}$ and feature matrix $X\in R^{2708\times 1433}$.Step 2: We employ Equations (Equation 1)–(Equation 5) as well as Algorithm 1 to generate $\tilde{A}\in R^{2708\times 2708}$.Step 3: Next, we use *A*, $\tilde{A}$ and *X* as input data for the shared convolution encoder, with a learned dimension of 128. As a result, we obtain the learned represention $H^{ori}\in R^{2708\times 128}$ and $H^{aug}\in R^{2708\times 128}$.Step 4: We take the learned representations $H^{ori}$ and $H^{aug}$ as inputs to two MLP encoders, namely, $g_{Node}$ and $g_{Comm}$. In $g_{Node}$, we set the output dimension to 32 and obtain the node-level representations $Z_{Node}^{ori}\in R^{2708\times 32}$ and $Z_{Node}^{aug}\in R^{2708\times 32}$. In $g_{Comm}$, the output dimension is set to the number of classes (which is 7 in this case), resulting in the community-level representations $Z_{Comm}^{ori}\in R^{2708\times 7}$ and $Z_{Comm}^{aug}\in R^{2708\times 7}$.Step 5: we use Equations (Equation 6) and (Equation 9) to compute the similarity of nodes and communities between original graph and augmentation view. We then use Equations (Equation 7), (Equation 8) and (Equation 10)–(Equation 13) to compute the node-level loss and community-level loss, respectively.Step 6: Finally, We use the Equation (Equation 14) to compute the overall loss, where the hyper-parameters α and β are set to 0.5. We update all the parameters according to Algorithm 2.Step 7: Once the training is complete, we take the original graph’s adjacency matrix *A* and feature matrix *X* as inputs to our model, resulting in the final community-level representation $Z_{Comm}^{ori}\in R^{2708\times 7}$. We then use the argmax function to assign nodes to communities based on their highest probability of membership.

#### 4.1.5. Baselines

We evaluate our method on six representative graph datasets and compare it with eight state-of-the-art graph algorithms. This includes four traditional methods: k-means [20], DeepWalk [24], GAE & VGAE [40], 6 existing graphs: GCA [32], MVGRL [33], DGI [31], HDI [41], gCoole [42].

### 4.2. Overall Performance

The experimental results presented in Table 2 demonstrate that our proposed method outperforms other state-of-the-art baselines in all six datasets in terms of NMI and ARI. Specifically, in the Amazon-Photo dataset, our method achieves more than 50% improvement in performance compared to the best baseline on Amazon-Computers and Coauthor-CS in terms of ARI. The remarkable results demonstrate the effectiveness of our proposed method in community detection. The combination of the community-level and node-level heads is the key to the performance improvement of our method. This will be further demonstrated in the ablation study presented in the following section.

### 4.3. Ablation Study

Three ablation studies are carried out to further understand the importance of data augmentation, the effect of two contrastive heads, and the reliance on the backbone network.

#### 4.3.1. Importance of Generator

We proposed a GAE-based data augmentation generator to enhance the performance of our contrastive learning method. As previous research has demonstrated, the success of contrastive learning heavily relies on the proper strategy of data augmentation [43]. To validate our approach, we evaluate our method on Cora using various data augmentation schemes, such as 10% EdgePerturb, 10% NodeDrop, 10% Subgraph and 10% AttrMask. In our experimental setup, the augmentation methods employed were as follows: 10% EdgePerturb: This method randomly drops each edge from the adjacency matrix with a probability of 0.1. 10% NodeDrop: With a probability of 0.1, this technique randomly removes individual nodes from the graph. 10% Subgraph: By selecting nodes with a probability of 0.1, a subgraph is constructed using the chosen nodes. And 10% AttrMask: With a probability of 0.1, this method masks node features by replacing them using 0. Figure 2 shows that our generator is more stable and produces more diverse and informative augmented views of the graph data. This is critical for contrastive learning since it requires a diverse set of views to learn meaningful representations of the underlying graph structure. Unlike traditional augmentation methods like edge perturbation and node dropping, which can randomly remove important nodes or edges and introduce uncertainty in community detection, our generator is more stable and preserves essential graph features in the produced views.

Among traditional augmentation methods, AttrMask performs the best, indicating that edge influence is stronger than attribute influence, and further highlighting the necessity of our proposed learnable augmentation based on edge reconstruction. Our learnable augmentation can adjust to the specific characteristics of the input graph data and generate views tailored to the task at hand. This is particularly important in real-world applications where the graph data can have varying characteristics and structures.

#### 4.3.2. Effect of Augmented-Level $\tau_{a}$

We investigate the impact of different levels of augmentation on community detection performance. To do so, we evaluate the NMI and ARI on Cora for augmentation levels ranging from $\tau_{a}=1$ to 5. In addition to NMI and ARI, we also use modularity Q as an evaluation metric, a common indicator for assessing community detection performance proposed by Newman [44]. The results are shown in Table 3. The predicted Q shown in Table 3 is a measure of the quality of the predicted community labels, calculated using the predicted labels and the real edges of the graph. The Original Q is calculated using the real labels and the real edges of the graph.

We conducted an evaluation of the modularity Q using both the real labels and the predicted labels. The result shows the following: (1) Our proposed method is effective in every level of augmented data. We found that when $\tau_{a}=3$, the number of preserved edges is more reasonable, and the result is optimal; (2) The predicted Q is higher than the original Q, indicating that our method is optimized towards making the community internally tight and externally sparse.

#### 4.3.3. Importance of Double-Contrastive Head

We conduct ablation studies on four datasets, Cora, Amazon-Photo, Amazon-Computers and Coauthor-CS, to demonstrate the effectiveness of our proposed node-level head (NCH) and community-level head (CCH). Specifically, we remove one of the two heads, and performed k-means in the node space for community assignments when the cluster-level contrastive head is removed. Results given in Table 4 demonstrate the complementary nature of the two heads and the improvement gained from their joint use. Notably, NCH performs better on Cora, while CCH performs better on the other three datasets. The results also demonstrate the importance and significance of considering the community level in community detection, as CCH can make direct community predictions.

To further validate the effectiveness of our approach, we use t-SNE [45] to visualize the node representation. As shown in Figure 3, when only applying the CCH, the distance among clusters is greater and the internal nodes of clusters are more compact. The NCH presents more blurred cluster boundaries and dispersed internal nodes. The ablation studies and visualization results demonstrate the innovation and importance of our proposed method in community detection.

## 5. Conclusions

In this paper, we present a novel self-supervised community detection algorithm based on graph contrastive learning, which improves current methods by designing new enhanced data generation strategies and a joint contrastive framework. Our algorithm presents several innovations.

Firstly, we introduce a novel data augmentation method that generates diverse views of networks by using a graph autoencoder and dot-product decoder. The augmented level provides more information about the network and enhances controllability. Our method can capture the complexity of community structure while including rich node information, effectively addressing the problem of poor robustness by traditional methods that randomly delete edges or nodes.

Secondly, we propose a joint contrastive framework that directly uses community structure information for community detection tasks, avoiding potential errors caused by two-step learning and achieving end-to-end learning. Community-level contrastive learning captures the global structure and topological features, while node-level contrastive learning considers the similarity and dissimilarity between nodes. By combining community-level and node-level contrastive learning, more in-depth information can be supplemented, improving the performance of community detection.

Most importantly, we extensively evaluate our method on multiple real-world graph datasets and compare it with other methods. The experimental results demonstrate the competitive performance of our algorithm in community detection tasks, proving its effectiveness and robustness.

Our proposed unsupervised community detection algorithm based on contrastive learning exhibits innovation in data augmentation and contrastive learning, and demonstrates excellent performance in experiments. Future research can further explore different data augmentation strategies and contrastive learning methods to improve the performance and applicability of community detection algorithms. Additionally, our method could be extended to handle larger-scale graphs and other related tasks such as link prediction, anomaly detection, and community evolution analysis.

## Figures and Tables

**Figure 1 entropy-25-00864-f001:**
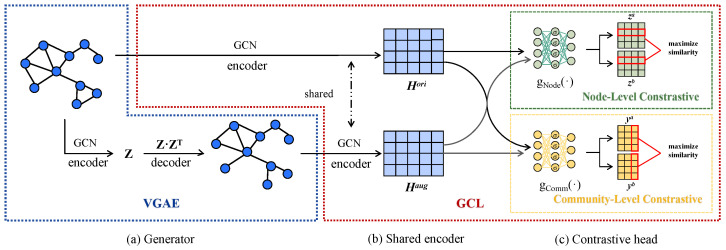
Framework overview of Community-CL model, which consist of (**a**) a augmentation generator, (**b**) a shared deep graph neural network, and (**c**) two contrastive heads which are designed for node-level and community-level, respectively.

**Figure 2 entropy-25-00864-f002:**
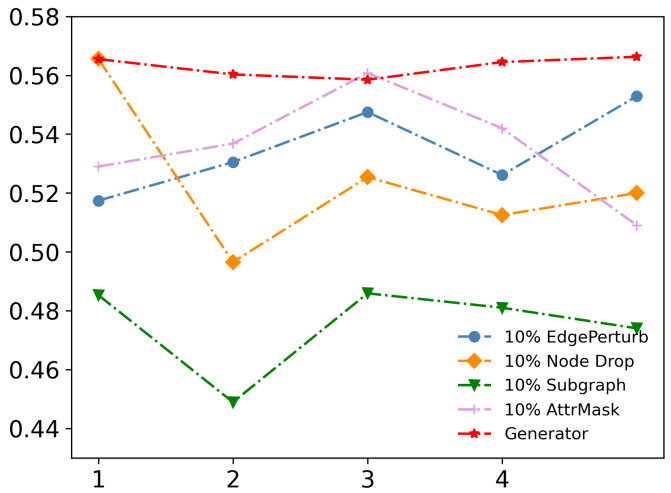
Ablation study on WikiCS by visualizing node representations with t-SNE.

**Figure 3 entropy-25-00864-f003:**
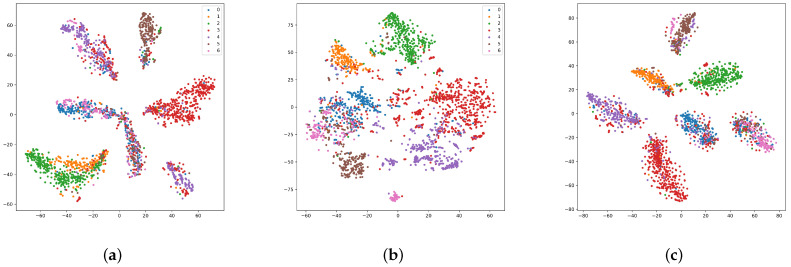
Ablation study on Cora by visualizing graph representations with t-SNE. (**a**) Community Contrastive Head. (**b**) Node Contrastive Head. (**c**) Community and Node Contrastive Heads.

**Table 1 entropy-25-00864-t001:** Statistics of datasets used for evaluations.

Dataset	Type	Nodes	Edges	Attributes	Classes
Cora	reference	2708	10,556	1433	7
Citeseer	reference	3327	9104	3703	6
Amazon-Photo	co-purchase	7487	119,043	745	8
Amazon-Computers	co-purchase	13,381	245,778	767	10
Coauthor-CS	co-author	18,333	81,894	6805	15
WikiCS	reference	11,701	216,123	300	10

**Table 2 entropy-25-00864-t002:** The clustering performance on six different size and type graphs. The best results are shown in boldface.

Dataset	Cora		Citeseer		Amazon-Photo		Amazon-Computers		Coauthor-CS		WikiCS	
**Metric**	**NMI**	**ARI**	**NMI**	**ARI**	**NMI**	**ARI**	**NMI**	**ARI**	**NMI**	**ARI**	**NMI**	**ARI**
k-means	0.167	0.229	0.17	0.27	0.235	0.112	0.192	0.086	0.498	0.315	0.244	0.022
DeepWalk	0.243	0.224	0.276	0.105	0.494	0.338	0.227	0.118	0.727	0.612	0.323	0.095
MVGRL	0.502	0.479	0.392	0.394	0.343	0.242	0. 244	0.141	0.733	0.637	0.254	0.101
DGI	0.498	0.447	0.378	0.381	0.365	0.253	0.318	0.165	0.754	0.639	0.309	0.130
HDI	0.449	0.352	0.350	0.341	0.430	0.310	0.347	0.216	0.725	0.616	0.240	0.104
GAE	0.389	0.293	0.174	0.141	0.614	0.493	0.441	0.258	0.727	0.613	0.241	0.094
VGAE	0.414	0.347	0.163	0.101	0.531	0.354	0.423	0.238	0.733	0.605	0.259	0.072
GCA	0.503	0.342	0.443	0.384	0.592	0.504	0.426	0.246	0.735	0.618	0.298	0.101
gCooLe	0.494	0.422	0.388	0.347	0.618	0.508	0.474	0.277	0.747	0.634	0.321	0.155
Our Method	**0.563**	**0.487**	**0.476**	**0.450**	**0.714**	**0.629**	**0.550**	**0.434**	**0.757**	**0.657**	**0.455**	**0.305**

**Table 3 entropy-25-00864-t003:** Effect of Augmented Level ($\tau_{a}$) on Community Detection Performance.

Augmented Level	NMI	ARI	Predicted Q	Original Q
**1**	0.533 ± 0.0327	0.465 ± 0.0753	0.734 ± 0.0082	0.6401
**2**	0.545 ± 0.0022	0.481 ± 0.0253	**0.743 ± 0.0095**	
**3**	**0.573 ± 0.0082**	**0.501 ± 0.0277**	0.735 ± 0.0127	
**4**	0.557 ± 0.0245	0.482 ± 0.0400	0.726 ± 0.0083	
**5**	0.558 ± 0.0107	0.486 ± 0.0294	0.717 ± 0.0150	

The optimal outcome is indicated by the bold part.

**Table 4 entropy-25-00864-t004:** Effect of two contrastive heads.

Dataset	Contrastive Head	NMI	ARI
Cora	Node+Community	**0.5368 + 0.018**	**0.4837 + 0.024**
	Node	0.4848 + 0.024	0.3847 + 0.045
	Community	0.4532 + 0.011	0.4051 + 0.015
Amazon-Photo	Node + Community	**0.7140 + 0.010**	**0.6293 + 0.008**
	Node	0.3824 + 0.030	0.2741 + 0.030
	Community	0.5752 + 0.000	0.5741 + 0.001
Amazon-Computers	Node + Community	**0.5506 + 0.0035**	**0.4338 + 0.0033**
	Node	0.4678 + 0.020	0.3053 + 0.032
	Community	0.4804 + 0.0014	0.3202 + 0.0020
Coauthor-CS	Node + Community	**0.7519 + 0.0132**	**0.6579 + 0.0108**
	Node	0.6432 + 0.0105	0.4357 + 0.0110
	Community	0.7243 + 0.0055	0.6378 + 0.0083

The optimal outcome is indicated by the bold part.

## Data Availability

Available on requests.

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
