# Peer review of "Community-CL: An Enhanced Community Detection Algorithm Based on Contrastive Learning"

_entropy, 2023, doi:10.3390/e25060864_

Round 1

Reviewer 1 Report

In this paper the authors propose Community Constrastive Learning, a framework for simultaneously learning node representations and detecting communities in a network.

The topic considered by the authors has been much studied in the past literature. However, the approach proposed in this paper appears innovative and could represent a new way for investigating this topic.

The analysis of related literature must be improved, especially with regard to community detection. For example, the authors should consider the approach described in the paper "Investigating community evolutions in TikTok dangerous and non-dangerous challenges" as well as many other approaches that have also been recently proposed in the literature on this topic.

The technical description of the approach is very good although in some places it is difficult to follow. I suggest the authors to include a leading example to make the proposed approach better understood.

The tests illustrated are interesting and seem to confirm the goodness of the proposed approach.

The paper's English seems good

Reviewer 2 Report

This work proposes the Community-CL framework for both learning node representations and community detection in a network. This paper will gain if the author will treat:

1. Row 152. Ref 29 is not new, as it has 4 years from publishing.

2. Row 165. the ref for GCC is not correctly compiled.

3. Fig. 1. Generator is written as Genator.

4. Row 205. n=|V|, not E.

5. The equations must be mentioned in the text.

6. Row 258 mentions the temperature parameter. It has to be explained before. Eq 6 TauN or TauNode, as written after? Who is j from the numerator?

7. Row 312 says that Alg 2 is the complete proposal, but the title of Alg 2 is "The framework of generator", the same as Alg 1.

8. Please give citations for all datasets used.

9. The implementation and the computational infrastructure are not described. Is your application available for further research?

10. Row 383. Please briefly describe the four methods used for augmentation.

11. The results in Sec 4.3.2 are for what dataset/s?

12. Ref 13 and 14 are identical.

Round 2

Reviewer 1 Report

The authors have striven to comply with my suggestion. Therefore, in my opinion the paper can be accepted

Reviewer 2 Report

The authors treated all the issues raised.